# Factors associated with COVID-19 vaccination schedule completion among adults in high-social-vulnerability neighborhoods in two Brazilian state capitals: A cross-sectional study

Fernanda Martins Iunes[1,2], Thaís Aranha Rossi[1,3], Fabiane Soares[3], Thiago S Torres[4], Valdilea G. Veloso[4], Débora Castanheira[4], Nathalia Suzart[1], Felipe Fagundes Soares[2], Alexandro Gesner Gomes dos Santos ![ORCID][2], Diana Zeballos[3], Ines Dourado[3], Laio Magno ![ORCID][1,3, 5]*

1 Departamento de Ciências da Vida, Universidade do Estado da Bahia, Salvador, Bahia, Brazil, 2 Prefeitura Municipal de Salvador, Secretaria Municipal de Saúde, Salvador, Bahia, Brazil, 3 Instituto de Saúde Coletiva, Universidade Federal da Bahia, Salvador, Bahia, Brazil, 4 Fundação Oswaldo Cruz, Instituto Nacional de Infectologia Evandro Chagas, Rio de Janeiro, Brazil, 5 Fundação Oswaldo Cruz, Instituto Gonçalo Moniz, Salvador, Bahia, Brazil

* laiomagnoss@gmail.com

## Abstract

This study aimed to examine the factors associated with Coronavirus disease 2019 (COVID-19) vaccination schedule completion among adults in socially vulnerable neighborhoods in two Brazilian state capitals. This cross-sectional study analyzed the data of individuals who attended primary healthcare (PHC) units in Salvador and Rio de Janeiro between July 2022 and May 2023. Data were collected using a structured sociobehavioral questionnaire. The outcome variable was COVID-19 vaccination schedule completion, defined as receiving ≥1 booster dose in the primary series (i.e., first and/or second dose or single-dose vaccine) plus one booster dose. The association between predictor variables and vaccination schedule completion was evaluated using multiple logistic regression. A total of 7,193 participants who reported having received ≥1 dose was included, 79.35% had completed the vaccination schedule. Participants aged >50 years (odds ratio [OR]=3.83; 95% confidence interval [CI]: 3.11–4.70), and cisgender women (OR=1.38; 95%CI: 1.20–1.57) and those with higher or graduate education (OR=2.44; 95%CI: 1.99–2.99), ≥2 comorbidities (OR=2.14; 95%CI: 1.56–2.92), mixed public and private healthcare use (OR=1.56; 95%CI: 1.30–1.89), a medical consultation in the last 12 months (OR=1.60; 95%CI: 1.38–1.85), and those who sought care at a primary health care unit in the last 12 months (OR=1.20; 95%CI: 1.03–1.41) had a higher odds of adherence to the scheduled vaccination than their respective counterparts. Conversely, individuals of Evangelical belief (OR=0.63; 95%CI: 0.56–0.72) and individuals residing in households with high occupancy density (OR=0.69; 95%CI: 0.57–0.82) showed lower vaccination completion than their respective counterparts. This study demonstrated

**Data availability statement:** Magno L. COVID-19 vaccination schedule completion among adult users of primary health care in Brazil. Harvard Dataverse; 2025. https://doi.org/10.7910/DVN/EIK1TM.

**Funding:** This project was made possible thanks to Unitaid's funding and support (#2017-15-FIOTECPrEP). Unitaid (https://unitaid.org/) accelerates access to innovative health products and lays the foundations for their scale-up by countries and partners. Unitaid is a hosted partnership of WHO. The funders had no role in study design, data collection and analysis, decision to publish, or preparation of the manuscript.

**Competing interests:** The authors have declared that no competing interests exist.

that sociodemographic and healthcare-related factors play a pivotal role in supporting COVID-19 vaccination adherence, contributing to higher coverage rates and reducing inequalities in vaccine access. Active outreach to unvaccinated individuals and strengthening primary healthcare are essential for increasing vaccination coverage and mitigating inequalities in access to health services.

## Introduction

The coronavirus disease 2019 (COVID-19) pandemic has triggered a global public health emergency, demanding a coordinated and robust response from healthcare systems worldwide. This crisis underscored the critical role of integrating primary healthcare (PHC) with other components of the Healthcare Network to effectively address emergency needs, both in terms of prevention and care of individuals affected by severe acute respiratory syndrome coronavirus 2.

In Brazil, PHC represents the central entry point to the Brazilian Health System (in Portuguese: *Sistema Único de Saúde–SUS)* [1], and its role involves organizing services and implementing actions that promote comprehensive health care for individuals and communities, addressing the impact of social determinants of health on the health-disease-care process [2]. As the first point of contact, PHC should be easily accessible [3] as an equitable, holistic, and comprehensive care so that individuals can use it whenever. In Brazil, PHC has extensive national coverage, allowing for a comprehensive clinical approach that considers the health conditions of the population in social, political, and economic dimensions. These PHC characteristics played a critical role in tackling COVID-19 by enabling responses adapted to local realities [4].

According to Donabedian [5], access to health services can be analyzed from two perspectives: geographical, which examines physical barriers to care, and socio-organizational, which investigates the administrative, social, and structural aspects that hinder access to adequate care. The combination of extensive operational reach and territorial knowledge allows effective responses to health crises.

Brazil was severely affected by the COVID-19 pandemic, ranking second worldwide in terms of the number of deaths in 2020, recording a peak of 4,148 daily deaths in April 2021 [6]. The country recorded 39,181,954 cases and 715,108 deaths by mid-February 2025, with a case fatality rate of 1.8% [7]. There was a 6% increase in cases (194,000 newly reported cases) and a 24% reduction in the number of deaths compared with the previous period, observed between November 11 and December 8, 2024 [7]. In this context, PHC plays an important role in reorganizing care flows, expanding access, and absorbing mild to moderate cases, contributing to the reduction of inequities in care provision [8].

Several countries began vaccine development research as early as 2020, considering the high transmissibility of COVID-19. In Brazil, the immunization process began on January 17, 2021, with the administration of CoronaVac following emergency use authorization by the Brazilian Health Regulatory Agency [9]. The National

Plan for Operationalization of COVID-19 Vaccination initially prioritized vulnerable groups, such as people aged 60 years or older, individuals living in long-term care facilities, and healthcare workers [10].

During the pandemic, the vaccines administered in Brazil were incorporated into the Ministry of Health's National Immunization Program (in Portuguese: *Programa Nacional de Imunizações–PNI*), which serves as a benchmark for international health organizations [11,12]. The following COVID-19 vaccines were included in this program at various stages of the pandemic: Pfizer/BioNTech (BNT162b2), AstraZeneca/Oxford (ChAdOx1), Janssen/Johnson and Johnson (Ad26.COV2.S), and CoronaVac/Sinovac Life Sciences [13].

Vaccination prevents an increase in severe cases by 74% according to studies and reduces mortality by 82% [14]. Despite advances in immunization, the political context in Brazil during the pandemic influenced vaccine hesitancy [15]. During the pandemic, the President of Brazil repeatedly questioned health measures and promoted disinformation about vaccines, claiming that they contained toxic substances and that natural immunity was more effective [16]. This narrative may have negatively influenced vaccination adherence, because the government's stance may have triggered insecurity and disinformation in the population. In fact, over 1.5 million Brazilians missed their second vaccine dose by April 2021 owing to unfounded fears, logistical difficulties, and confusion regarding dose intervals [9].

Considering the lack of information on the factors influencing the completion of COVID-19 vaccination schedules among socially vulnerable populations in Brazil, this study aimed to analyze the factors associated with COVID-19 vaccination schedule completion among adults in high-vulnerability neighborhoods in two Brazilian state capitals.

## Materials and methods

### Study design and population

This cross-sectional study was part of a larger project titled "TQT-COVID-19 Study." The study population included individuals of all ages seeking COVID-19 testing and who met the following inclusion criteria: i) Residents of the Cabula-Beirú Health District in Salvador-BA or the Manguinhos neighborhood in Rio de Janeiro-RJ and ii) Presence of symptoms associated with COVID-19 that began between 3 and 7 days after disease onset and/or reporting close contact with a confirmed COVID-19 case (if asymptomatic, between 5 and 7 days after the last contact). Details of this study are provided elsewhere [17]. For this analysis, we included adults who received at least one dose of a COVID-19 vaccine.

### Setting

The TQT-COVID-19 Study was conducted at 17 PHC units in the Cabula-Beirú Health District, Salvador (Bahia state, Northeast Brazil), and two units in Manguinhos, Rio de Janeiro (Rio de Janeiro state, Southeast Brazil). The Cabula-Beirú District in Salvador has a population of 392,542 inhabitants spread across an area of 25.89 km$^2$ [18]. Data in Salvador were collected between July 4, 2022, and March 3, 2023. Manguinhos, a neighborhood located in the northern zone of Rio de Janeiro spreads across 261.84 hectares with an estimated population of 40,586 [19]. In Rio de Janeiro, data collection periods varied by unit: one unit collected data from November 8, 2022, to May 16, 2023, and the other from December 19, 2022, to May 16, 2023.

### Data collection and instrument

Data were collected using a structured sociobehavioral questionnaire administered by trained researchers for standardized application. The previously trained research team consisted of health professionals, community health workers (CHWs), and researchers.

Participants were asked to answer a structured socio-behavioral questionnaire while awaiting their test results. The questionnaire included thematic blocks addressing sociodemographic, clinical, and health service access and utilization

as well as behavioral data. Individuals aged 12 years or older were invited to answer a structured questionnaire; for children under 12 years, the questions were answered by their parents or guardians.

## Study variables

The outcome variable was COVID-19 vaccination schedule completion, categorized as follows:

i)   Incomplete schedule: the participant reported receiving only the first or second dose or a single-dose vaccine.

ii)  Complete schedule: the participant reported receiving the primary series (i.e., first and second doses or single-dose vaccine) plus at least one booster dose.

In Fig 1, the timeline of COVID-19 vaccination in Brazil is illustrated. At the time of data collection, COVID-19 vaccines were only available at PHC units. The following vaccines were being administered: CoronaVac, Pfizer, Oxford/AstraZeneca, and Janssen, with the latter being a single-dose vaccine.

The following potential associated variables were analyzed:

i)   Sociodemographic characteristics: Age (18–30, 31–50, > 50 years), sex (cisgender man, cisgender woman, transgender people), race/skin color (White, Brown, Black, Indigenous, Asian were grouped in Black/Brown versus others), education level (up to elementary/middle school, high school complete or incomplete, higher education, and graduate), Evangelical religion (no, yes), and monthly family income were categorized using two minimum wages as a cut-off. In 2022 the minimum wage in Brazil was BRL 1,212 (approximately USD 232, based on the 2022 average exchange rate of 1 USD ≈ 5.22 BRL), and density of people per room was calculated by dividing the number of individuals living in the household by the total number of rooms. This variable was categorized using a cut-off point of 0.5: low density (<0.5), indicating better living conditions, and high density (≥0.5), indicating poorer living conditions.

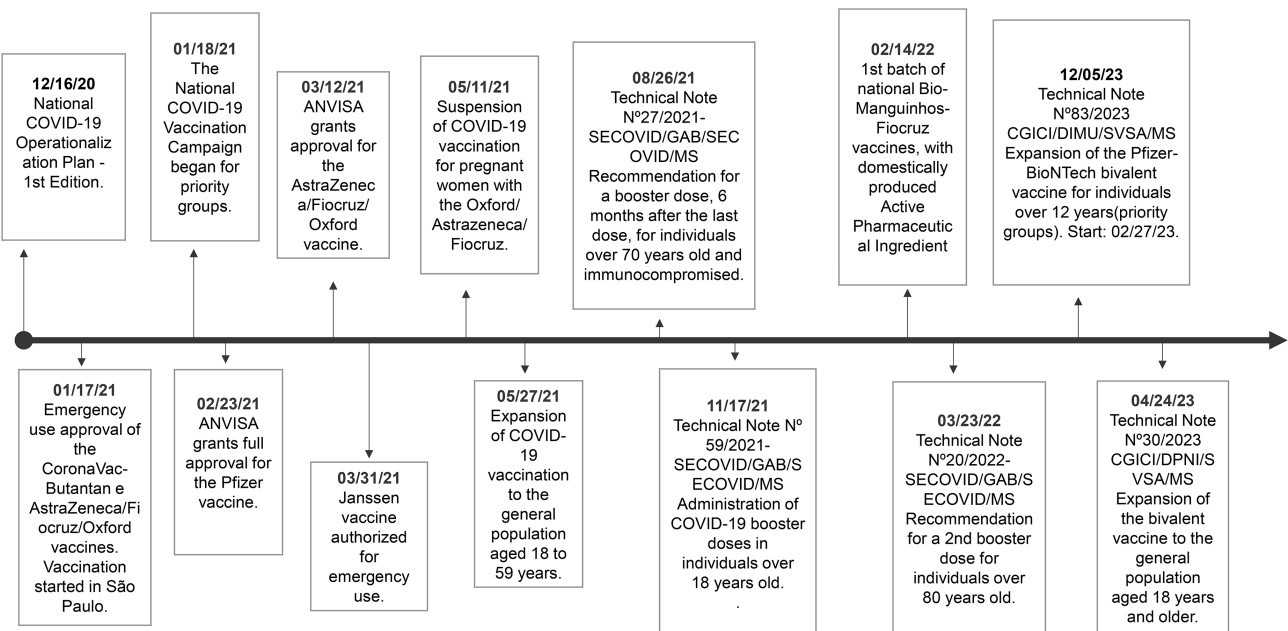

**Fig 1. Timeline of COVID-19 schedule implementation in Brazil.**

ii) Clinical aspects: Participants were asked to self-report diagnosed chronic comorbidities with a predefined list of specific chronic conditions by answering "yes" or "no" to each: obesity, diabetes mellitus, heart disease or high blood pressure, respiratory diseases (e.g., asthma, pulmonary emphysema, tuberculosis), current or past cancer treatment, hematological diseases (including sickle cell anemia), advanced-stage chronic kidney disease, chromosomal conditions associated with immune fragility (e.g., Down syndrome, Turner syndrome), liver diseases (e.g., fatty liver, hepatitis, cirrhosis), autoimmune diseases (e.g., systemic lupus erythematosus, rheumatoid arthritis, immune thrombocytopenia), immunodeficiencies (e.g., HIV infection, leukemia), or were asked to specify other conditions. The participants were then classified as having none, one comorbidity, or two or more comorbidities. Self-perceived risk of contracting COVID-19 (none, low, medium, or high).

iii) Access to health services: Forms of access to health services (exclusively through SUS (i.e., public), private, public, and private), household registration at a Family Health Unit (FHU) (yes, no, do not know), last medical consultations (in the last 12 months more than 12 months or never), sought care at the health unit in the last 12 months (yes, and was successful; no or unable to secure an appointment/slot), and reporting of any type of discrimination in the health service (no, yes).

## Data analysis

Individuals younger than 18 years were excluded from the analysis because of different vaccination schedules in the PNI for children and adolescents, which would have complicated the combined analysis. A descriptive analysis was conducted to characterize the study population. The association between predictor variables and vaccination schedule completion was investigated by bivariate analysis using the chi-square test, and multivariate analysis using multiple logistic regression with estimates of adjusted odds ratios (aORs) and 95% confidence intervals (95%CIs). Variables with a p-value ≤0.05 in the bivariate analysis were included in the multivariable model. Participants with missing values for the covariates included in the multivariate analysis were excluded from the final model. The final model was adjusted using a backward elimination procedure, removing variables stepwise while comparing models based on the Akaike Information Criterion, Bayesian Information Criterion, and likelihood ratio tests to identify the best-fitting model. Multicollinearity was assessed using variance inflation factors (VIF). The goodness of fit of the final model was evaluated using the Hosmer-Lemeshow test, and its discriminative ability was assessed using the area under the receiver operating characteristic curve. We also evaluated the potential hierarchical structure of the data by comparing the logistic regression model with a multilevel logistic regression model including UBS as a random intercept. The multilevel approach was fitted to account for possible within-UBS clustering of the outcome. All analyses were conducted using STATA software version 16.1.

## Ethics approval and consent to participate

This study was conducted in accordance with the guidelines of Ethical Research Resolution No. 466/2012 concerning research involving humans in Brazil. This project was granted approval by the Research Ethics Committees of the World Health Organization (protocol identification: CERC.0128A and CERC.0128B) and local Research Ethics Committees (protocol identification in Salvador, Institute of Collective Health: No. 53844121.4.1001.5030; and Rio de Janeiro–Evandro Chagas National Institute of Infectious Diseases: No. 53844121.4.2001.5262, National School of Public Health: No. 53844121.4.3001.5240, and Municipal Health Secretariat of Rio de Janeiro: No. 53844121.4.3002.5279). In the TQT-COVID-19 Study, all participants aged 18 years or older signed an informed consent form; for minors, the consent form was signed by the parents, and those capable of reading and understanding also signed the Assent Form.

## Results

A total of 12,286 participants were registered in the TQT-COVID-19 Study, and 7,193 adults who received at least one dose of any COVID-19 vaccine were included (Fig 2).

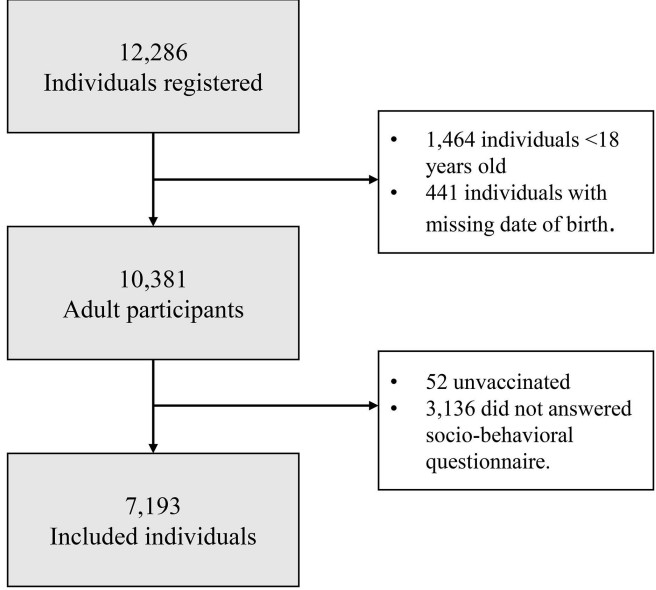

**Fig 2. Flowchart of the study population inclusion.**

Among the 7,193 adult participants, most had a complete schedule (79.35%), 2,628 (36.53%) received at least one booster, and 3,080 (42.82%) received two boosters. Among those with incomplete schedules, a small proportion did not complete the primary series (n = 176; 2.45%) (Fig 3).

Most participants resided in Salvador (76.64%), were cisgender women (70.26%), had Black or Brown race/skin color (84.58%), were aged between 31 and 50 years (43.26%), and had a complete or incomplete high school education (49.59%). Additionally, 35.51% reported being Evangelical, most participants preferred not to respond to income questions, and 27.37% reported an income between BRL 1,212.01 and BRL 2,424.00 (Table 1).

Most respondents resided in households with medium or high density (64.09%) and perceived their risk of contracting COVID-19 as moderate or high (72.65%). Most participants (67.22%) reported no comorbidities. With respect to health service utilization, 68.26% accessed services exclusively through the *SUS,* and 63.18% had household registration with an FHU. In the 12 months preceding the survey, 78.67% reported having had at least one medical consultation, 79.49% had sought care at a health unit in the last 12 months and were attended to, and 6.18% reported experiencing discrimination within health services (Table 1).

In the multivariate analysis (Table 2), older participants were significantly more likely to complete the COVID-19 vaccination schedule than younger adults. Individuals aged 31–50 years had 49% higher odds of completion (OR=1.49; 95%CI: 1.30–1.71), and those older than 50 years had nearly four times higher odds (OR=3.83; 95%CI: 3.11–4.70) than participants aged 18–30 years. Cisgender women also had 38% higher odds of completing the vaccination than cisgender men (OR=1.38; 95%CI: 1.20–1.57). Higher educational attainment was strongly associated with completion: participants with complete/incomplete high school had 37% (OR=1.37; 95%CI: 1.17–1.60) higher odds, and those with higher education or graduate degrees were more than twice as likely to have completed the vaccination schedule (OR=2.44; 95%CI: 1.99–2.99) compared with those with up to elementary or middle school education. In contrast, identifying with an Evangelical religion was associated with lower odds of completion (OR=0.63; 95%CI: 0.56–0.72). Participants living in more crowded households (≥1 person per room) were less likely to complete the vaccination program (OR=0.69; 95%CI: 0.57–0.82). Comorbidities increased the likelihood of vaccination, particularly in those with two or more conditions (OR=2.14;

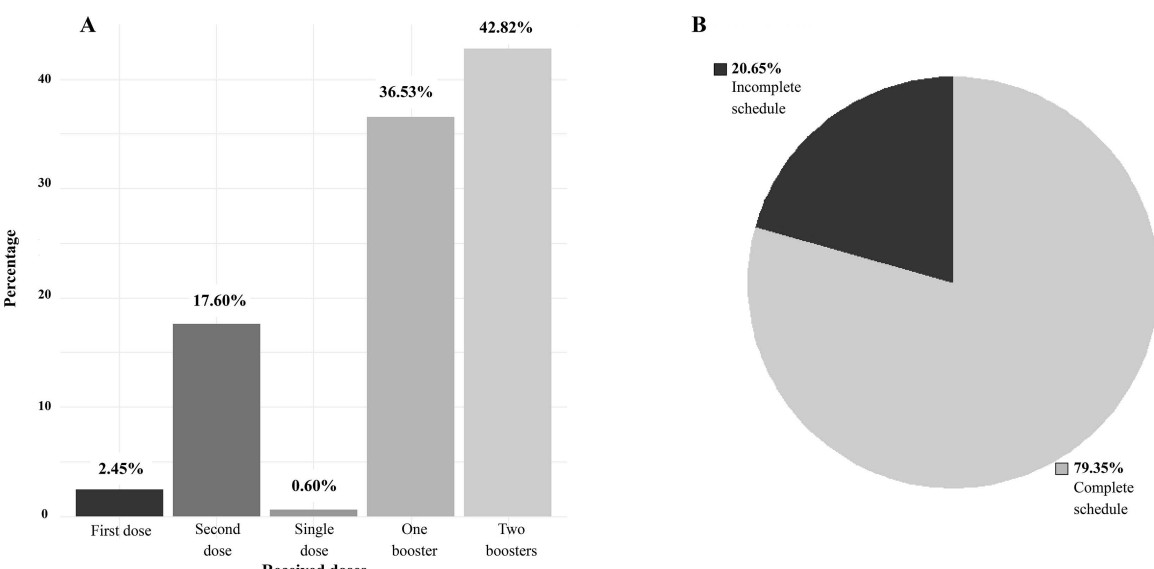

**Fig 3. Number of doses received and COVID-19 vaccination status among adults participating in the TQT-COVID-19 Study.** (A) Percentage of participants by number of doses received. (B) Distribution of participants by vaccination status. The incomplete schedule includes the primary series (i.e., first dose and/or second dose, or a single-dose vaccine). The complete schedule refers to the primary series plus at least one booster dose.

95%CI: 1.56–2.92). Finally, greater access to and use of health services were positively associated with vaccination: individuals using both public and private services (OR=1.56; 95%CI: 1.30–1.89), those who had a medical consultation within the past 12 months (OR=1.60; 95%CI: 1.38–1.85) and those who sought care at a primary health care unit in the last 12 months and were able to get it (OR=1.20; 95%CI: 1.03–1.41), were more likely to have completed the vaccination schedule. All variables presented VIF values below 2 (mean VIF = 1.31), indicating the absence of relevant multicollinearity. The multilevel analysis (S1 Table) showed a very low intraclass correlation coefficient (ICC = 1.2%), indicating minimal clustering of the outcome at the UBS level. Additionally, odds ratios and corresponding confidence intervals were nearly identical between the multilevel and standard logistic regression models, with no substantive changes in the interpretation of the results. Therefore, the standard logistic regression model was retained, as the main conclusions remained unchanged.

## Discussion

The findings highlight important sociodemographic and healthcare-related determinants of completing the COVID-19 vaccination schedule. We identified a higher proportion of COVID-19 vaccination schedule completion among those with greater engagement in health services. A higher vaccination completion rate was also associated with older age, female cisgender identity, and higher educational attainment. Conversely, lower completion rates among younger participants, those identifying with the Evangelical religion, and individuals living in crowded households highlight social and structural barriers that may have influenced vaccination uptake.

High vaccination adherence among older individuals aligns with the policies prioritizing this group in the National Plan for the Operationalization of COVID-19 Vaccination. According to national data, in February 2022, 60% of the population aged 60 years or older had received the COVID-19 booster dose, reflecting the effectiveness of the strategy aimed at protecting this vulnerable group [8]. Immunization has become an essential tool for preventing severe infections and mortality associated with vaccine-preventable diseases with aging [20]. Modeling studies indicate that the implementation of COVID-19 vaccination in Brazil had a substantial impact on reducing hospitalizations and deaths among older adults.

**Table 1. Sociodemographic, clinical, and health service characteristics, overall and by COVID-19 vaccination schedule completion, among adults participating in the TQT-COVID-19 Study, Brazil, 2022–2023.**

| Variables | N | n (%) | COVID-19 vaccination schedule completion | | p-value |
|---|---|---|---|---|---|
| | | | Yes (n = 5,708; 79.35%) | No (n = 1,485; 20.6%) | |
| **Sociodemographic characteristics, n (%)** | | | | | |
| **Study city** | 7,193 | | | | 0.002 |
| Salvador | | 5,513 (76.64) | 4,420 (80.17) | 1,093 (19.83) | |
| Rio de Janeiro | | 1,680 (23.36) | 1,288 (76.67) | 392 (23.33) | |
| **Age** | 7,193 | | | | <0.001 |
| 18–30 years | | 2,109 (29.32) | 1,477 (70.03) | 632 (29.97) | |
| 31–50 years | | 3,112 (43.26) | 2,439 (78.37) | 673 (21.63) | |
| >50 years | | 1,972 (27.42) | 1,792 (90.87) | 180 (9.13) | |
| **Gender identity** | 7,153 | | | | <0.001 |
| Cisgender man | | 2,103 (29.40) | 1,583 (75.27) | 520 (24.73) | |
| Cisgender woman | | 5,026 (70.26) | 4,075 (81.08) | 951 (18.92) | |
| Transgender individuals* | | 24 (0.34) | – | – | |
| **Race/skin color** | 7,186 | | | | 0.091 |
| White/Indigenous/Asian | | 1,108 (15.42) | 900 (81.23) | 208 (18.77) | |
| Brown/Black | | 6,078 (84.58) | 4,801 (78.99) | 1,277 (21.01) | |
| **Education level** | 7,183 | | | | <0.001 |
| Up to elementary/middle school | | 1,674 (23.31) | 1,270 (75.87) | 404 (24.13) | |
| Complete or incomplete high school | | 3,562 (49.59) | 2,736 (76.81) | 826 (23.19) | |
| Higher education and graduate | | 1,947 (27.11) | 1,694 (87.01) | 253 (12.99) | |
| **Evangelical religion** | 6,891 | | | | <0.001 |
| No | | 4,444 (64.49) | 3,662 (82.40) | 782 (17.60) | |
| Yes | | 2,447 (35.51) | 1,821 (74.42) | 626 (25.58) | |
| **Income**** | 7,190 | | | | <0.001 |
| Government aid or <BRL 1,212.00 | | 1,268 (17.64) | 945 (74.53) | 323 (25.47) | |
| BRL 1,212.01–BRL 2,424.00 | | 1,968 (27.37) | 1,533 (77.90) | 435 (22.10) | |
| BRL 2,424.01–BRL 6,060.00 | | 1,040 (14.46) | 847 (81.44) | 193 (18.56) | |
| >BRL 6,060.01 | | 257 (3.57) | 232 (90.27) | 25 (9.73) | |
| Did not answer | | 2,657 (36.95) | 2,151 (80.96) | 506 (19.04) | |
| **Density of people per room** | 7,190 | | | | <0.001 |
| Low density (≤0.5) | | 2,582 (35.91) | 2,160 (83.66) | 422 (16.34) | |
| Medium density (0.5–0.99) | | 3,486 (48.48) | 2,769 (79.43) | 717 (20.57) | |
| High density (>=1) | | 1,122 (15.61) | 779 (69.43) | 343 (30.57) | |
| **Clinical aspects, n (%)** | | | | | |
| **Number of comorbidities** | 7,193 | | | | <0.001 |
| None | | 4,835 (67.22) | 3,655 (75.59) | 1,180 (24.41) | |
| Only one | | 1,709 (23.76) | 1,457 (85.25) | 252 (14.75) | |
| Two or more | | 649 (9.02) | 596 (91.83) | 53 (8.17) | |
| **Self-perceived risk of getting COVID-19** | 7,187 | | | | 0.349 |
| None or low | | 1,966 (27.35) | 1,546 (78.64) | 420 (21.36) | |
| Medium or high | | 5,221 (72.65) | 4,158 (79.64) | 1,063 (20.63) | |
| **Access to health services, n (%)** | | | | | |
| **Forms of access to health services** | 7,183 | | | | <0.001 |
| Exclusively through SUS (Public) | | 4,903 (68.26) | 3,725 (75.97) | 1,178 (24.03) | |

*(Continued)*

**Table 1.** (Continued)

| Variables | N | n (%) | COVID-19 vaccination schedule completion | | p-value |
|---|---|---|---|---|---|
| | | | **Yes (n = 5,708; 79.35%)** | **No (n = 1,485; 20.6%)** | |
| Private | | 872 (12.14) | 747 (85.67) | 125 (14.33) | |
| Public and private | | 1,408 (19.60) | 1,231 (87.43) | 177 (12.57) | |
| **Household registered at an FHU** | 7,186 | | | | 0.272 |
| Yes | | 4,540 (63.18) | 3,592 (79.12) | 948 (20.88) | |
| No | | 2,038 (28.36) | 1,640 (80.47) | 398 (19.53) | |
| Don't know | | 608 (8.46) | 473 (77.80) | 135 (22.20) | |
| **Last medical consultation** | 7,193 | | | | <0.001 |
| More than 12 months or never | | 1,534 (21.33) | 1,070 (69.75) | 464 (30.25) | |
| In the last 12 months | | 5,659 (78.67) | 4,638 (81.96) | 1,021 (18.04) | |
| **Sought care at the health unit in the last 12 months** | 7,193 | | | | <0.001 |
| No/appointment could not be scheduled | | 1,475 (20.51) | 1,107 (75.05) | 368 (24.95) | |
| Yes, and received care | | 5,718 (79.49) | 4,601 (80.47) | 1,117 (19.53) | |
| **Report of discrimination in health services** | 7,190 | | | | 0.079 |
| No | | 6,746 (93.82) | 5,341 (79.17) | 1,405 (20.83) | |
| Yes | | 444 (6.18) | 367 (82.66) | 77 (17.34) | |

\* Transgender individuals were excluded from the bivariate and multivariate analyses because of the small sample size. \*\* Income was excluded from bivariate and multivariate analysis because of large missing values equivalent to 36.95% of the participants.

SUS: Brazilian National Health System; FHU: Family Health Unit.

Between January and August 2021, it is estimated that more than 167,000 COVID-19 hospitalizations among individuals aged 60 years or older were averted, and during the same period, vaccination may have saved more than 58,000 lives of older adults [21]. These data emphasize the effectiveness of vaccines in reducing morbidity and mortality associated with COVID-19.

Another prioritized population was individuals with comorbidities and noncommunicable diseases (NCDs) owing to their increased vulnerability to COVID-19, which increases disease severity, hospital stay, and mortality risk [22]. Individuals with NCDs tend to use health services more frequently because of their continuous need for medical care. Consequently, the increased use of services may explain the higher vaccination rates among these patients [23]. The pandemic may have further exacerbated this situation by reducing access to elective consultations for individuals with NCDs [24,25].

Studies indicate that women show higher vaccination adherence than men; for instance, a study conducted in Brazil revealed that women had less vaccine hesitancy than men, despite men having a higher risk of clinical worsening of the disease [26]. Vasconcelos et al. highlighted that factors such as risk perception and self-care positively influence this discrepancy in vaccine adherence between the sexes [27]. In this regard, women express higher concern for disease prevention and risk perception, which contributes to higher vaccination rates [28].

Educational level is also a relevant factor for vaccination adherence. Silva et al. documented that individuals with higher educational levels tend to have lower vaccine hesitancy, owing to greater access to trustworthy scientific information and less exposure to disinformation [29]. Likewise, Edwards et al. demonstrated that individuals with higher education are more likely to trust science and reject conspiracy theories about vaccination [30].

The relationship between household crowding and socioeconomic status is well documented in scientific literature, with evidence suggesting that larger families often face considerable economic challenges. These challenges are often associated with precarious housing conditions and limited access to essential resources [31]. Monteiro and Veras reported that housing is a high-cost commodity with selective access that disproportionately excludes economically disadvantaged

**Table 2. Multivariate analysis of TQT-COVID-19 Study adult population variables associated with COVID-19 vaccination schedule completion in two Brazilian state capitals, 2022–2023. (N = 6,827).**

| Variables | OR (95%CI) |
|---|---|
| **Age** | |
| 18–30 years | 1 |
| 31–50 years | 1.49 (1.30–1.71) |
| >50 years | 3.83 (3.11–4.70) |
| **Gender identity** | |
| Cisgender man | 1 |
| Cisgender woman | 1.38 (1.20–1.57) |
| **Education level** | |
| Up to elementary/middle school | 1 |
| Complete or incomplete high school | 1.37 (1.17–1.60) |
| Higher education and graduate | 2.44 (1.99–2.99) |
| **Evangelical religion** | |
| No | 1 |
| Yes | 0.63 (0.56–0.72) |
| **Density of people per room** | |
| ≤0.5 | 1 |
| 0.5–0.99 | 1.01 (0.88–1,17) |
| >=1 | 0.69 (0.57–0.82) |
| **Number of comorbidities** | |
| None | 1 |
| Only one | 1.33 (1.13–1.57) |
| Two or more | 2.14 (1.56–2.92) |
| **Forms of access to health services** | |
| Exclusively through SUS (Public) | 1 |
| Private | 1.34 (1.07–1.67) |
| Public and private | 1.56 (1.30–1.89) |
| **Last medical consultation** | |
| More than 12 months or never | 1 |
| In the last 12 months | 1.60 (1.38–1.85) |
| **Sought care at the health unit in the last 12 months** | |
| No/appointment could not be scheduled | 1 |
| Yes, and received care | 1.20 (1.03–1.41) |

Goodness of fit p=0.28 and AUC=0.71.

SUS: Brazilian National Health System; FHU: Family Health Unit.

groups [32]. Such findings underscore the critical need for public policies aimed at improving housing and the broader socioeconomic conditions of larger families. Such policies are essential for fostering social inclusion and ensuring equitable access to the basic services necessary for a dignified life. Crawshaw et al. found that hesitancy was more evident in regions with greater social inequality [33]. In Brazil, the vaccine uptake intention is notably lower in regions with increased social inequality [34].

Vaccine hesitancy is a complex concept, and religion played an important role in vaccination completion among the study participants. Studies have shown that religious beliefs and disinformation are determining factors for vaccine

denial [15]. The dissemination of false news was another critical factor in reducing vaccination adherence, as reported by Gonçalves et al. [15,33].

This study had limitations that should be acknowledged. First, its cross-sectional design does not allow the establishment of causal relationships between the factors analyzed and the completion of the COVID-19 vaccination schedule. Second, the type of vaccine received, which could have influenced schedule completion owing to misinformation or preferences, was not adequately collected, preventing a more detailed analysis of this aspect. In addition, income, an important socioeconomic indicator, had a high proportion of missing data and was therefore excluded from the bivariate and multivariate analyses. However, other variables that reflect socioeconomic conditions, such as educational level, household crowding, and access to private health services, were included and consistently indicated that individuals with greater socioeconomic disadvantages were less likely to complete the vaccination schedule. Furthermore, external validity may be limited, as the study was based on a non-probabilistic sample recruited through health services. Consequently, caution is warranted when generalizing the results beyond similar service-based populations. Despite these limitations, the study has notable strengths, including its large sample size, ability to assess the relationship between healthcare engagement and vaccination completion, use of a standardized questionnaire, and employment of trained interviewers.

## Conclusions

This study demonstrated that sociodemographic and healthcare-related factors play a pivotal role in supporting adherence to COVID-19 vaccination, contributing to higher coverage rates and reducing inequalities in vaccine access. Individuals in continuous contact with the healthcare system exhibited higher rates of vaccination schedule completion. Proactive outreach by Brazil's PHC teams was fundamental in facilitating vaccination access. Moreover, age, sex, and educational attainment significantly influenced adherence, highlighting the need for targeted policies for vaccine-hesitant groups.

The implementation of educational initiatives, in conjunction with active outreach and home visits by CHWs, could further improve vaccination uptake. Strengthening PHC, along with clear and evidence-based communication, is essential for addressing vaccine hesitancy and advancing health equity. In conclusion, PHC should continue to be reinforced as a strategic pillar of the national immunization policy to ensure broad, equitable, and effective access to vaccination, both for COVID-19 and future public health emergencies. Investments in PHC, health education, and strategies to combat disinformation are crucial for increasing vaccination coverage and decreasing inequities in access to vaccines.

## Supporting information

**S1 Table. Multilevel logistic regression analysis of factors associated with COVID-19 vaccination schedule completion among adults in two Brazilian state capitals, 2022–2023 (TQT-COVID-19 Study; N = 6,827).**
(DOCX)

## Author contributions

**Conceptualization:** Thaís Aranha Rossi, Thiago S Torres, Valdilea G Veloso, Ines Dourado, Laio Magno.

**Data curation:** Fernanda Martins Iunes.

**Formal analysis:** Fernanda Martins Iunes, Diana Zeballos, Laio Magno.

**Funding acquisition:** Valdilea G Veloso, Ines Dourado, Laio Magno.

**Investigation:** Thaís Aranha Rossi, Fabiane Soares, Débora Castanheira, Nathalia Suzart, Felipe Fagundes Soares, Alexandro Gesner Gomes dos Santos.

**Methodology:** Fabiane Soares.

**Project administration:** Fabiane Soares, Thiago S Torres, Nathalia Suzart, Ines Dourado, Laio Magno.

**Supervision:** Thaís Aranha Rossi, Débora Castanheira, Nathalia Suzart.

**Validation:** Thiago S Torres.

**Writing – original draft:** Fernanda Martins Iunes, Laio Magno.

**Writing – review & editing:** Fernanda Martins Iunes, Thaís Aranha Rossi, Fabiane Soares, Thiago S Torres, Valdilea G Veloso, Débora Castanheira, Nathalia Suzart, Felipe Fagundes Soares, Alexandro Gesner Gomes dos Santos, Diana Zeballos, Ines Dourado, Laio Magno.

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
