## [Decision Letter · Decision Letter 0]

25 Sep 2025

Dear Dr. Magno

**In addition to the reviewers’ comments, please see my additional observations below.**

We look forward to receiving your revised manuscript.

Kind regards,

Vinícius Silva Belo

Academic Editor

PLOS ONE

**Journal Requirements:**

1. When submitting your revision, we need you to address these additional requirements. Please ensure that your manuscript meets PLOS ONE's style requirements, including those for file naming. The PLOS ONE style templates can be found at https://journals.plos.org/plosone/s/file?id=wjVg/PLOSOne_formatting_sample_main_body.pdf and https://journals.plos.org/plosone/s/file?id=ba62/PLOSOne_formatting_sample_title_authors_affiliations.pdf 2. Thank you for stating the following financial disclosure: This project was made possible thanks to Unitaid's funding and support (#2017-15-FIOTECPrEP). Unitaid (https://unitaid.org/) accelerates access to innovative health products and lays the foundations for their scale-upby countries and partners. Unitaid is a hosted partnership of WHO.   Please state what role the funders took in the study.  If the funders had no role, please state: "The funders had no role in study design, data collection and analysis, decision to publish, or preparation of the manuscript." If this statement is not correct you must amend it as needed. Please include this amended Role of Funder statement in your cover letter; we will change the online submission form on your behalf. 3. When completing the data availability statement of the submission form, you indicated that you will make your data available on acceptance. We strongly recommend all authors decide on a data sharing plan before acceptance, as the process can be lengthy and hold up publication timelines. Please note that, though access restrictions are acceptable now, your entire data will need to be made freely accessible if your manuscript is accepted for publication. This policy applies to all data except where public deposition would breach compliance with the protocol approved by your research ethics board. If you are unable to adhere to our open data policy, please kindly revise your statement to explain your reasoning and we will seek the editor's input on an exemption. Please be assured that, once you have provided your new statement, the assessment of your exemption will not hold up the peer review process. 4. Please amend either the abstract on the online submission form (via Edit Submission) or the abstract in the manuscript so that they are identical. 5. If the reviewer comments include a recommendation to cite specific previously published works, please review and evaluate these publications to determine whether they are relevant and should be cited. There is no requirement to cite these works unless the editor has indicated otherwise. 

**Additional Editor Comments:**

Given the hierarchical structure of the data across health units, it is necessary analyze the data using multilevel models, as standard logistic regression may underestimate standard errors and overlook variability between units. In addition, the potential for collinearity among the modeled variables should be formally assessed and, if present, corrected. A discussion on the statistical power of variables with small numbers of participants is also required, as well as a more in-depth discussion on the external validity of the findings.

Reviewers' comments:

**Comments to the Author**

1. Is the manuscript technically sound, and do the data support the conclusions?

Reviewer #1: Yes

Reviewer #2: Yes

2. Has the statistical analysis been performed appropriately and rigorously?

Reviewer #1: Yes

Reviewer #2: Yes

3. Have the authors made all data underlying the findings in their manuscript fully available?

Reviewer #1: Yes

Reviewer #2: Yes

4. Is the manuscript presented in an intelligible fashion and written in standard English?

Reviewer #1: Yes

Reviewer #2: Yes

**Reviewer #1:** The manuscript addresses an important topic and provides valuable insights regarding vaccination uptake in socially vulnerable populations. However, several methodological and presentation issues require careful revision to strengthen the validity and clarity of the findings.The manuscript addresses an important topic and provides valuable insights regarding vaccination uptake in socially vulnerable populations. However, several methodological and presentation issues require careful revision to strengthen the validity and clarity of the findings.

Minor Methodological Limitations

Definition of the outcome

The study defined a complete vaccination schedule as the primary series plus two booster doses. However, during the data collection period, not all participants were eligible for two boosters (depending on age, risk group, and national guidelines). This may have led to misclassification of individuals as “incomplete,” even if they were up to date according to contemporaneous recommendations. Requiring two boosters for all participants risks classifying as “incomplete” those who were not eligible at the time (due to age, risk profile, interval since last dose, or guideline changes). As a reflection, the outcome should be redefined as “adherence according to eligibility” (age/conditions + guidelines in force at the date of interview) and analyses repeated.

Inadequate handling of missing data

Table 4 reports N=6,805, lower than the total of vaccinated participants with at least one dose (7,193), suggesting a complete-case analysis without description of missing-data handling. The absence of a clear strategy for managing missing values may introduce systematic bias. It is recommended to report the pattern and proportion of missing data by variable and conduct sensitivity analyses.

Inconsistencies and Critical Data Presentation Issues

Reversed percentages in Table 3 header: listed as “Yes (n=3,080, 57.2%) / No (n=4,113, 42.8%).” However, 3,080/7,193=42.82%, which is consistent with the distribution in Table 1 (“up to second booster” = 42.82%). The header percentages must be corrected.

Inconsistent numeric notation: mixed use of separators (e.g., “28,52” and “2.882” in Table 3), which may confuse readers and hinder verification. All tables should be standardized—using a dot for decimals and consistent thousand separators. The problem recurs across tables. Recommendation: apply a consistent numeric format (e.g., decimal point; thousand separator as comma or space).

Key corrections and additions include:

1. Correct the Table 3 header (reversed percentages) and standardize numeric notation across all tables.

2. Redefine the outcome based on eligibility criteria and reanalyze, retaining the current definition as a sensitivity analysis.

3. Address missing data (describe patterns; consider multiple imputation; include income with “not reported” as a category). Report the analyzed sample size (N) for each model.

4. Include a timeline of booster recommendations (by age and risk group), indicating the proportion of participants eligible in each stratum.

**Reviewer #2:**  General Assessment General Assessment

The manuscript addresses an important subject that is the factors associated with COVID-19 vaccination schedule completion in socially vulnerable neighborhoods in two Brazilian capitals. The study benefits from a large sample size, standardized data collection, and the inclusion of two distinct settings. However, several clarifications, additions, and structural adjustments would strengthen the manuscript and make the findings more transparent.

Major Points

Study Population and Flow Diagram

Although the questionnaire was reportedly administered to individuals aged ≥12 years, the analysis only includes participants aged ≥18 years. Please clarify this point. How many were excluded? I recommend including a flow diagram showing the number of individuals invited, excluded (<18 years) and retained for analysis. In addition, consider including in the supplementary material the full questionnaire used in data collection.

Study Setting

The manuscript lacks a dedicated paragraph in the Methods section describing the study setting. I recommend that the authors clearly describe the health care units selected for data collection as well as the municipalities and neighborhoods included in the analysis. Providing this information would allow readers to better understand the context, representativeness and potential differences between the study sites.

Vaccine Type, Timing and Adherence

The data collection instrument reportedly captured vaccine type. In Brazil, multiple vaccine schemes were implemented depending on age, comorbidities and city. Different vaccines were subject to distinct public perceptions and adverse event profiles, for example controversy surrounding CoronaVac and Pfizer and reports of post vaccination side effects. Please consider either analyzing or at least discussing whether vaccine type may have influenced completion of the second dose or booster uptake.

If vaccination dates for the study population were recorded, please make this explicit in the manuscript. If these dates are available, it is important to indicate whether all participants had equal opportunity to complete the vaccination schedule during the study period. For example, older adults and individuals with comorbidities may have had more time to receive boosters than younger adults. Likewise, individuals who began their vaccination scheme close to the time of questionnaire administration might not have had the opportunity to complete the schedule. Please place the timing of vaccination of the study population in the context of the national and local vaccination campaigns. This context helps explain some of the observed associations. It would also be useful to comment on possible differences in campaign implementation between Salvador and Rio de Janeiro, given the higher completion rate in Rio.

Use of SUS vs. Private Health Services

The title of the manuscript implies that that the study population are only adult users of primary health care. Does part of the study population exclusively use health insurance or private services? If so, the title may not accurately describe the total sample. Please clarify this point. My uncertainty comes from the following description: “Access to health services: Forms of access to health services (exclusively through SUS, health insurance and private, all services).” Please explain clearly what “all services” means. Also, the journal has an international audience, it would be helpful to describe item (iii) more explicitly as there are terms that only Brazilian readers would easily understand.

Also, Table 4 shows that exclusive SUS users had lower odds of completing the vaccination schedule than those with private, if I clearly understood, or mixed public–private use (all services??). The Discussion should explicitly acknowledge this distinction and avoid overstating PHC’s role beyond what the data support, as already indicated in the first sentence of the Conclusion. The statement “The study demonstrated that PHC plays a pivotal role…” implies causality. A more accurate formulation would be: “Our findings indicate that identifying PHC as the usual source of care is associated with higher odds of completing the vaccination schedule, suggesting that PHC may facilitate adherence; however, exclusive use of SUS services showed lower completion rates.”

Data Visualization

The manuscript relies heavily on tables and would benefit from more intuitive graphics. I suggest including a forest plot displaying adjusted odds ratios from the multivariable model. Also, maybe a stacked bar chart showing vaccination stages by city, age group, sex and religion,

Additionally, Table 2 appears redundant because Table 3 already presents counts and percentages in the context of the analysis. If the authors wish to retain the total numbers, they could add the total N and percentage in a third column rather than keeping a separate table.

Discussion Structure

I recommend restructuring the Discussion to begin with a succinct paragraph summarizing the main quantitative findings (odds ratios). It is also very important to include a dedicated paragraph on the limitations of the study, such as the self-reported nature of vaccination data. If no analysis by vaccine type is presented, this should be acknowledged as a limitation as well. The cross-sectional design should also be mentioned because it precludes causal inference. In addition, the Discussion could emphasize the strengths of the study, such as the large sample of PHC users in high vulnerability areas, the use of a standardized questionnaire and the employment of trained interviewers.

**Do you want your identity to be public for this peer review?** For information about this choice, including consent withdrawal, please see our For information about this choice, including consent withdrawal, please see our Privacy Policy .

Reviewer #1: **Yes:** Everton Ferreira LemosEverton Ferreira Lemos

Reviewer #2: No

While revising your submission, please upload your figure files to the Preflight Analysis and Conversion Engine (PACE) digital diagnostic tool, https://pacev2.apexcovantage.com/ . PACE helps ensure that figures meet PLOS requirements. To use PACE, you must first register as a user. Registration is free. Then, login and navigate to the UPLOAD tab, where you will find detailed instructions on how to use the tool. If you encounter any issues or have any questions when using PACE, please email PLOS at . PACE helps ensure that figures meet PLOS requirements. To use PACE, you must first register as a user. Registration is free. Then, login and navigate to the UPLOAD tab, where you will find detailed instructions on how to use the tool. If you encounter any issues or have any questions when using PACE, please email PLOS at figures@plos.org . Please note that Supporting Information files do not need this step.. Please note that Supporting Information files do not need this step.

---

## [Author Response · Author response to Decision Letter 1]

17 Dec 2025

Dear reviewers:

We would like to thank you for the opportunity to resubmit a revised copy of our manuscript “Factors associated with COVID-19 vaccination schedule completion among adults in high-social-vulnerability neighborhoods in two Brazilian state capitals: A cross-sectional study”. We would also like to express our gratitude to the reviewers for their feedback and helpful comments regarding corrections or modifications.

Please find below our answers to the reviewers' comments:

Reviewer #1: The manuscript addresses an important topic and provides valuable insights regarding vaccination uptake in socially vulnerable populations. However, several methodological and presentation issues require careful revision to strengthen the validity and clarity of the findings.

Minor Methodological Limitations

Definition of the outcome

The study defined a complete vaccination schedule as the primary series plus two booster doses. However, during the data collection period, not all participants were eligible for two boosters (depending on age, risk group, and national guidelines). This may have led to misclassification of individuals as “incomplete,” even if they were up to date according to contemporaneous recommendations. Requiring two boosters for all participants risks classifying as “incomplete” those who were not eligible at the time (due to age, risk profile, interval since last dose, or guideline changes). As a reflection, the outcome should be redefined as “adherence according to eligibility” (age/conditions + guidelines in force at the date of interview) and analyses repeated.

Answer: Thank you for these comments. We included a figure with the COVID-19 vaccination timeline in Brazil, describing the start of the campaign and the progressive inclusion of different groups. For this reason, we analyzed the data considering a complete vaccination schedule as the primary series plus at least one booster dose, rather than two boosters, because during the data collection period, all participants aged 18 years or older were eligible for the primary series and at least one booster dose.

Inadequate handling of missing data

Table 4 reports N=6,805, lower than the total of vaccinated participants with at least one dose (7,193), suggesting a complete-case analysis without description of missing-data handling. The absence of a clear strategy for managing missing values may introduce systematic bias. It is recommended to report the pattern and proportion of missing data by variable and conduct sensitivity analyses.

Answer: Thank you for this suggestion. In Table 1 we included the number of participants with complete information for each variable. We excluded participants with missing information, as stated in the Methods section.

Inconsistencies and Critical Data Presentation Issues

Reversed percentages in Table 3 header: listed as “Yes (n=3,080, 57.2%) / No (n=4,113, 42.8%).” However, 3,080/7,193=42.82%, which is consistent with the distribution in Table 1 (“up to second booster” = 42.82%). The header percentages have been corrected accordingly.

Answer: We revised it accordingly.

Inconsistent numeric notation: mixed use of separators (e.g., “28,52” and “2.882” in Table 3), which may confuse readers and hinder verification. All tables should be standardized—using a dot for decimals and consistent thousand separators. The problem recurs across tables. Recommendation: apply a consistent numeric format (e.g., decimal point; thousand separator as comma or space).

Answer: We revised it accordingly.

Key corrections and additions include:

1. Correct the Table 3 header (reversed percentages) and standardize numeric notation across all tables.

Answer: We have made the corrections.

2. Redefine the outcome based on eligibility criteria and reanalyze, retaining the current definition as a sensitivity analysis.

Answer: We redefined our outcome variables based on the availability criteria and reanalyzed the data.

3. Address missing data (describe patterns; consider multiple imputation; include income with “not reported” as a category). The analyzed sample size (N) for each model is reported.

Answer: We included the total sample size (N) for each variable and model in the tables and included the missing data in the income variable as a limitation; however, we had other variables to assess socioeconomic factors.

4. Include a timeline of booster recommendations (by age and risk group), indicating the proportion of participants eligible in each stratum.

Answer: We included a figure showing the COVID-19 vaccination timeline for Brazil. Based on this timeline, individuals aged 18 years and older were already eligible to receive a booster dose before the study’s data collection period. Therefore, we considered the vaccination schedule to be complete for individuals receiving one or two booster doses and did not find it necessary to include the proportion of participants eligible for each stratum.

Reviewer #2: General Assessment

The manuscript addresses an important subject that is the factors associated with COVID-19 vaccination schedule completion in socially vulnerable neighborhoods in two Brazilian capitals. The study benefits from a large sample size, standardized data collection, and the inclusion of two distinct settings. However, several clarifications, additions, and structural adjustments would strengthen the manuscript and make the findings more transparent.

Major Points

Study Population and Flow Diagram

Although the questionnaire was reportedly administered to individuals aged ≥12 years, the analysis only includes participants aged ≥18 years. Please clarify this point. How many were excluded? I recommend including a flow diagram showing the number of individuals invited, excluded (<18 years) and retained for analysis. In addition, consider including in the supplementary material the full questionnaire used in data collection.

Answer: Thank you for your comments and suggestions. We included a flow chart showing the number of individuals invited, excluded, and retained in the analysis. Individuals under 18 years of age were excluded because they were not eligible for the full vaccination schedule. The complete questionnaire used in the study has been added as an appendix to the publication of the protocol (Reference 18 in our article).

Study Setting

The manuscript lacks a dedicated paragraph in the Methods section describing the study setting. I recommend that the authors clearly describe the healthcare units selected for data collection as well as the municipalities and neighborhoods included in the analysis. Providing this information would allow readers to better understand the context, representativeness and potential differences between the study sites.

Answer: We included information describing the study setting and health care units selected for data collection, which are also available in the protocol published in reference 18.

Vaccine Type, Timing and Adherence

The data collection instrument reportedly captured vaccine type. In Brazil, multiple vaccine schemes were implemented depending on age, comorbidities and city. Different vaccines were subject to distinct public perceptions and adverse event profiles, for example controversy surrounding CoronaVac and Pfizer and reports of post vaccination side effects. Please consider either analyzing or at least discussing whether vaccine type may have influenced completion of the second dose or booster uptake.

Answer: We collected information on vaccine type; however, this variable was subject to considerable recall bias and missing data, as it was self-reported rather than obtained from vaccination cards. We did not collect information regarding the reasons for not completing the vaccination schedule; however, we agree that this information would have been valuable. We have added this information in the Discussion.

If vaccination dates for the study population were recorded, please make this explicit in the manuscript. If these dates are available, it is important to indicate whether all participants had equal opportunity to complete the vaccination schedule during the study period. For example, older adults and individuals with comorbidities may have had more time to receive boosters than younger adults. Likewise, individuals who began their vaccination scheme close to the time of questionnaire administration might not have had the opportunity to complete the schedule. Please place the timing of vaccination of the study population in the context of the national and local vaccination campaigns. This context helps explain some of the observed associations. It would also be useful to comment on possible differences in campaign implementation between Salvador and Rio de Janeiro, given the higher completion rate in Rio.

Answer: The variables containing the dates of the vaccines received by the participants also had many missing values, making it impossible to perform the analyses. To improve the visualization of the eligibility of the population aged 18 years and older for COVID-19 vaccination, we added a figure showing the COVID-19 vaccination timeline in Brazil, describing the start of vaccination and the inclusion of different groups. For this reason, we reanalyzed the data considering a complete vaccination schedule as the primary series plus at least one booster dose, rather than two boosters, because during the data collection period, all participants aged 18 years or older were eligible for the primary series and at least one booster dose. We have also added a brief description of the Salvador and Rio de Janeiro municipalities to the Methods section .

Use of SUS vs. Private Health Services

The title of the manuscript implies that that the study population are only adult users of primary health care. Does part of the study population exclusively use health insurance or private services? If so, the title may not accurately describe the total sample. Please clarify this point. My uncertainty comes from the following description: “Access to health services: Forms of access to health services (exclusively through SUS, health insurance and private, all services).” Please explain clearly what “all services” means. Also, the journal has an international audience, it would be helpful to describe item (iii) more explicitly as there are terms that only Brazilian readers would easily understand.

Answer: We agree that not all the participants were users of primary healthcare services. Therefore, we updated the title to: “Factors associated with COVID-19 vaccination schedule completion among adults in high-social-vulnerability neighborhoods in two Brazilian state capitals: A cross-sectional study.” We also redefined the variable “forms of access to health services” to clarify that “all services” refer to both public and private services. In addition, we provided a clearer explanation of item (iii) to an international audience.

Also, Table 4 shows that exclusive SUS users had lower odds of completing the vaccination schedule than those with private, if I clearly understood, or mixed public–private use (all services??). The Discussion should explicitly acknowledge this distinction and avoid overstating PHC’s role beyond what the data support, as already indicated in the first sentence of the Conclusion. The statement “The study demonstrated that PHC plays a pivotal role…” implies causality. A more accurate formulation would be: “Our findings indicate that identifying PHC as the usual source of care is associated with higher odds of completing the vaccination schedule, suggesting that PHC may facilitate adherence; however, exclusive use of SUS services showed lower completion rates.”

Answer: We have accepted your suggestions and revised the Discussion section accordingly. We explicitly acknowledge the distinction between exclusive SUS users and those using private or public and private services. We have rephrased the statement about PHC to reflect associations rather than causality, as recommended.

Data Visualization

The manuscript relies heavily on tables and would benefit from more intuitive graphics. I suggest including a forest plot displaying adjusted odds ratios from the multivariable model. Also, maybe a stacked bar chart showing vaccination stages by city, age group, sex and religion,

Additionally, Table 2 appears redundant because Table 3 already presents counts and percentages in the context of the analysis. If the authors wish to retain the total numbers, they could add the total N and percentage in a third column rather than keeping a separate table.

Answer: We have included this figure in place of Table 1. We also included sample sizes for each variable.

Discussion Structure

I recommend restructuring the Discussion to begin with a succinct paragraph summarizing the main quantitative findings (odds ratios). It is also very important to include a dedicated paragraph on the limitations of the study, such as the self-reported nature of vaccination data. If no analysis by vaccine type is presented, this should be acknowledged as a limitation as well. The cross-sectional design should also be mentioned because it precludes causal inference. In addition, the Discussion could emphasize the strengths of the study, such as the large sample of PHC users in high vulnerability areas, the use of a standardized questionnaire and the employment of trained interviewers.

Answer: We have revised the Discussion accordingly.

---

## [Decision Letter · Decision Letter 1]

26 Jan 2026

Dear Dr. Magno,

We look forward to receiving your revised manuscript.

Kind regards,

Vinícius Silva Belo

Academic Editor

PLOS One

Journal Requirements:

**Additional Editor Comments:**

Although the reviewers considered the authors’ responses to most comments satisfactory, an important methodological point raised in the first round of review was not addressed. The following comment, which was also provided in the initial evaluation, should be carefully considered and explicitly addressed in the revised manuscript:

*Given the hierarchical structure of the data across health units, it is necessary analyze the data using multilevel models, as standard logistic regression may underestimate standard errors and overlook variability between units. In addition, the potential for collinearity among the modeled variables should be formally assessed and, if present, corrected. A discussion on the statistical power of variables with small numbers of participants is also required, as well as a more in-depth discussion on the external validity of the findings.*

Reviewers' comments:

Reviewer's Responses to Questions

**Comments to the Author**

Reviewer #1: All comments have been addressed

Reviewer #2: All comments have been addressed

2. Is the manuscript technically sound, and do the data support the conclusions?

Reviewer #1: Yes

Reviewer #2: Yes

3. Has the statistical analysis been performed appropriately and rigorously?

Reviewer #1: Yes

Reviewer #2: Yes

4. Have the authors made all data underlying the findings in their manuscript fully available?

Reviewer #1: Yes

Reviewer #2: Yes

5. Is the manuscript presented in an intelligible fashion and written in standard English?

Reviewer #1: Yes

Reviewer #2: Yes

Reviewer #1: The revised manuscript shows substantial improvement and adequately addresses the main concerns raised during the review process. The inclusion of the vaccination timeline figure improves transparency. The Methods section is clearer, particularly regarding the study setting, population flow, and analytical strategy. Overall, the study presents a robust and relevant contribution to the literature on COVID-19 vaccination adherence in socially vulnerable settings.

Reviewer #2: The authors have carefully implemented substantial revisions throughout the manuscript, addressing the comments and concerns raised during the previous review round. The revised version shows clear improvements in clarity, methodological description, data presentation, and overall coherence. As a result, I have no further comments and consider the manuscript suitable to proceed in the editorial process.”

**Do you want your identity to be public for this peer review?** For information about this choice, including consent withdrawal, please see our For information about this choice, including consent withdrawal, please see our Privacy Policy .

Reviewer #1: **Yes:** Everton Ferreira LemosEverton Ferreira Lemos

Reviewer #2: No

---

## [Author Response · Author response to Decision Letter 2]

9 Mar 2026

Comment

Although the reviewers considered the authors’ responses to most comments satisfactory, an important methodological point raised in the first round of review was not addressed. The following comment, which was also provided in the initial evaluation, should be carefully considered and explicitly addressed in the revised manuscript:

Given the hierarchical structure of the data across health units, it is necessary analyze the data using multilevel models, as standard logistic regression may underestimate standard errors and overlook variability between units. In addition, the potential for collinearity among the modeled variables should be formally assessed and, if present, corrected. A discussion on the statistical power of variables with small numbers of participants is also required, as well as a more in-depth discussion on the external validity of the findings.

Response

Dear Editor,

We sincerely apologize for having inadvertently overlooked this important methodological comment raised during the first round of review. We appreciate your careful attention in bringing this issue back to our notice.

Thank you for the suggestion to consider a multilevel modeling approach. We carefully evaluated a random-effects logistic regression including UBS as a random intercept and compared its performance with the standard logistic regression model.

The intraclass correlation coefficient was very low (ICC = 1.2%), indicating minimal clustering of the outcome at the UBS level. In addition, odds ratios and confidence intervals were nearly identical across both models, and the substantive interpretation of the results did not change. We consider that the added complexity of the multilevel model did not provide meaningful analytical gains; for that reason, and because the core conclusions remain unchanged, we therefore chose to retain the standard logistic regression model. We included a supplementary Table S1 as a Supplementary Material, and this information in the results: “The multilevel analysis (S1 Table) showed a very low intraclass correlation coefficient (ICC = 1.2%), indicating minimal clustering of the outcome at the UBS level. Additionally, odds ratios and corresponding confidence intervals were nearly identical between the multilevel and standard logistic regression models, with no substantive changes in the interpretation of the results. Therefore, the standard logistic regression model was retained, as the main conclusions remained unchanged“.

Multicollinearity was assessed using variance inflation factors (VIF). All variables presented VIF values below 2 (mean VIF = 1.31), indicating the absence of relevant multicollinearity. We included this information in the results: “All variables presented VIF values below 2 (mean VIF = 1.31), indicating the absence of relevant multicollinearity”.

Given the large overall sample size and the high expected cell counts across the analyzed variables, the study has adequate statistical power to detect meaningful associations, and the risk of unstable estimates due to sparse data is minimal. We included the following information in the limitations: “Furthermore, external validity may be limited, as the study was based on a non-probabilistic sample recruited through health services. Consequently, caution is warranted when generalizing the results beyond similar service-based populations”.

---

## [Editor Report · Decision Letter 2]

15 Mar 2026

Factors associated with COVID-19 vaccination schedule completion among adults in high-social-vulnerability neighborhoods in two Brazilian state capitals: A cross-sectional study

PONE-D-25-35448R2

Dear Dr. Magno,

We’re pleased to inform you that your manuscript has been judged scientifically suitable for publication and will be formally accepted for publication once it meets all outstanding technical requirements.

Kind regards,

Vinícius Silva Belo

Academic Editor

PLOS One

Additional Editor Comments (optional):

Congratulations!
---

## [Editor Report · Acceptance letter]

PONE-D-25-35448R2

PLOS One

Dear Dr. Magno,

I'm pleased to inform you that your manuscript has been deemed suitable for publication in PLOS One. Congratulations! Your manuscript is now being handed over to our production team.

Kind regards,

on behalf of

Dr. Vinícius Silva Belo

Academic Editor

PLOS One